# A Literature Review of Unintentional Intoxications of Nonhuman Primates

**DOI:** 10.3390/ani12070854

**Published:** 2022-03-29

**Authors:** Jaco Bakker, Arieh Bomzon

**Affiliations:** 1Biomedical Primate Research Center (BPRC), 2288GJ Rijswijk, The Netherlands; 2Consulwrite Editorial and Consulvet Laboratory Animal Consultancies, Pardess Hanna-Karkur, Haifa 3707426, Israel; arieh@consulwrite.com

**Keywords:** chemical hazards, intoxication, nonhuman primates, poisoning

## Abstract

**Simple Summary:**

This report is an overview of the published veterinary literature on unintentional poisonings in wild and captive nonhuman primates (NHP). Selected intoxications will be described with emphasis on the clinical signs, laboratory, and main postmortem findings as well as the available treatment options. Mostly, NHP died suddenly and unexpectedly without showing any preceding clinical signs. The (presumptive) diagnosis of a poisoning was mostly made postmortem by laboratory testing of post-mortem fluid, tissue samples, and stomach contents. From these reports, we concluded that the toxic threats to wild NHP are different to those of captive NHP because of the environment in which they live. We also concluded that a poisoning or an intoxication should be part of a differential diagnosis when a NHP presents with clinical signs that cannot be linked to a known disorder or dies suddenly with no preceding clinical signs.

**Abstract:**

Reports of unintentional intoxications in nonhuman primates (NHP) are few and an up-to-date review of such intoxications in NHP is lacking. We reviewed the published veterinary literature on unintentional intoxications in wild and captive NHP in order to provide a useful resource on known toxic agents of NHP for veterinarians, caregivers, and researchers who work with NHP. To these ends, we first conducted a literature search for books, book chapters, peer-reviewed publications, conference proceedings, and newsletters in academic literature databases such as Google Scholar, PubMed, BioOne Complete, and Web of Science using the words and word combinations such as heavy metals, pesticides, poisonings, and nonhuman primates. We then evaluated the search results for those reports that we considered as clinically relevant and then classified them according to the toxic agent. We identified lead, zinc, phytotoxins, pesticides, botulinum toxin, polychlorinated biphenyls, and snake and spider venoms as the main toxic agents in wild and captive NHP. We concluded that the toxic threats to wild NHP are different to those of captive NHP because of the environment in which they live. We recommend that an intoxication should be part of a differential diagnosis when a NHP presents with clinical signs that cannot be linked to a known disorder or dies suddenly with no preceding clinical signs. In cases of the former, laboratory testing for known toxins and pollutants should be conducted. In cases of the latter, a thorough postmortem examination, together with extensive laboratory testing, for known toxins and environmental pollutants in all tissues and organs should be performed.

## 1. Introduction

Wild and captive nonhuman primates (NHP) are exposed and potentially vulnerable to many natural and man-made toxic threats. Nevertheless, wild NHP are capable of coping with these threats using strategies, namely avoidance, dilution, gastrointestinal degradation, or detoxification, which require genetic potential, learning from parents and conspecifics in their social group, or prior experience through random food sampling and experimentation. Captive NHP are also at high risk for intoxications when they are often housed in an outdoor enclosure in a vivarium or zoo that is in or close to a large urban and industrial city. These NHP are potentially exposed to urban-industrial air pollution due to industrial and vehicle exhausts, waste incineration, and the domestic and industrial use of petroleum-based products, cleaners, pesticides, and paints, amongst others. 

Many guidelines on the housing of captive NHP have been published. According to the guidelines of the United Kingdom’s National Center for the Replacement, Refinement, and Reduction of Animals in Research (NC3Rs), an ideal housing system for NHP should provide ample space and complexity to allow NHP to engage daily in a normal range of social and non-social behaviors. Where possible, enclosure furniture should be made from organic materials without chemical preservatives. In addition, a housing facility for NHP should be kept in good repair, protect the animals from injury, and be constructed of materials that can be easily cleaned and sanitized or removed or replaced when worn or soiled.

Most reports about intoxications in captive and wild NHP are often published as case reports, which have a narrow focus. Additionally, the historical reports on unintentional and inadvertent poisonings in NHP are not always digitally available and some are only available online as abstracts. Hence, an up-to-date review of such poisonings in NHP is lacking. In this report, we communicate our findings after mining the published veterinary literature on unintentional intoxications in NHP in order to provide a useful resource on known toxic agents of NHP for veterinarians, caregivers, and researchers who work with NHP. To these ends, we first carried out a literature search for books, book chapters, peer-reviewed publications, conference proceedings, and newsletters in academic literature databases such as Google Scholar, PubMed, BioOne Complete, and Web of Science by using words and word combinations, such as heavy metals, pesticides, poisonings, and nonhuman primates, in order to identify potentially relevant publications. We then evaluated reports that we considered as clinically relevant and then classified them into eight groups according to the types of toxic agent (Table 1). In this review, we describe the main toxic threats to NHP as well as the clinical signs, diagnosis, and treatments of these poisonings. We have not included reports on toxic disorders or the adverse effects of experimental intoxications that were induced in NHP models of human disease. 

## 2. Lead

Lead is a naturally occurring metal that is found in small amounts in rocks and soils and is an abundant environmental contaminant. Until leaded motor vehicle fuels were phased out in the mid- to late-1990s, air emissions of lead from the transportation sector (vehicle exhausts) were the major contributors of atmospheric lead. Since lead is also used industrially in the production of ceramic products, paints, metal alloys, and batteries, lead emissions into the atmosphere nowadays are associated with these industrial operations. These air emissions can either be inhaled (dust inhalation) or the lead can be ingested after settling from the air; such ingestion is considered as the main route of exposure to lead. The amounts of lead in the environmental air, food, and water supply of captive NHP are small, do not cause lead poisoning, and are comparable to those that most humans in urban populations are exposed [1]. Notwithstanding, lead-based paints are still widely used in many low- and middle-income developing countries across Asia, Africa, Latin America, and Europe, and the production and trade of lead-based paints is still widespread globally [2].

The health risk associated with lead exposure and ingestion is well documented. After its adsorption, lead distributes throughout the body in the blood, where it adversely affects its oxygen-carrying capacity, and accumulates in the bones because it approximates calcium and other bone-seeking elements. Depending on the level and duration of exposure, lead can adversely affect the renal function, and the nervous, hematological, gastrointestinal, cardiovascular, musculoskeletal, immune, reproductive, and developmental systems [3,4,5,6,7,8,9,10]. In NHP, the clinical signs of a hematological and a central nervous system disorder should invoke suspicion of lead poisoning. It should be noted that some captive NHP may be predisposed to ingesting cage paint because of their foraging habits. It should also be noted that social stress and teething in immature captive NHP are associated with the gnawing of cage bars.

Although a diagnosis of lead poisoning in NHP can be based on history and clinical and laboratory findings, most diagnoses are usually made postmortem. The results of postmortem examinations on NHP from the taxa *Lemuroidea*, *Simioidea*, and *Catarhiniat* revealed that the NHP died from lead poisoning following ingestion of a lead-containing cage paint [3,4,5,6]. Prior to their death, the NHP usually exhibited non-specific signs of ill health, such as partial anorexia and lethargy. Although vomiting and diarrhea were seen in several NHP, others exhibited no clinical signs. Other reported clinical signs of lead poisoning include weakness, pallor of the mucous membranes, and jaundice. The reported neurological signs of lead poisoning in NHP for which histories were available include convulsions, which are typical of acute amaurotic epilepsy, transient paralysis, paresis, apparent blindness (vision loss), tremors, and ataxia. Laboratory testing of NHP with suspected lead poisoning revealed elevated blood lead levels and anemia with immature and stippled erythrocytes [7,8,9] and basophils [10]. The main postmortem findings are lesions of lead encephalopathy, acid-fast intranuclear inclusion bodies in renal epithelia or hepatocytes, excess lead in the liver, metaphyseal bone changes (lead lines in the bones of immature monkeys), and necrosis of striated muscle fibers [3,4,5,6].

Treating lead poisoning in NHP comprises removal of the lead source(s) and oral chelation therapy with either calcium disodium edetate (EDTA), 2,3-dimercapto-1-propanol (BAL), D-penicillamine, or 2,3-dimercaptosuccinic acid at the manufacturer’s recommended dose for humans. Of these agents, 2,3-dimercaptosuccinic acid is often selected as the therapeutic agent because its adverse effects are minimal [11,12].

## 3. Zinc

Zinc is an essential trace element, ubiquitous in nature, and is biologically necessary for the function of many metalloenzymes and metalloproteins. It is dietary sourced from meats, eggs, dairy products, nuts, seeds, legumes, and cereals. After ingestion, the low pH of the stomach causes the release of free zinc. About 25–50% of ingested zinc is absorbed in the small intestine and this absorption is homeostatically controlled. Following its absorption, it accumulates in the liver, kidneys, pancreas, and spleen. Most absorbed zinc is excreted in the feces via biliary secretions. In cells, zinc participates directly in catalysis (metalloenzymes) and contributes to maintaining protein structure and stability (metalloproteins). 

Galvanization is the process of applying a protective zinc coat to iron or steel in order to prevent corrosion and rusting. Zinc toxicosis mainly occurs in captive NHP and is usually the result of licking the galvanized metal of their cages or enclosures or the ingestion of galvanized nuts and bolts from their metal cages or enclosures [13,14,15]. Most of the toxic effects of zinc occur when free zinc is released and zinc salts such as zinc chloride are formed in the stomach’s acidic environment. These salts irritate and cause corrosion of the gastric and intestinal (duodenal) mucosa. Accordingly, the first clinical signs of zinc toxicosis, often vomiting, anorexia, and weight loss, are exhibited a few days after zinc ingestion. Prolonged zinc exposure has a deleterious effect on almost every organ system and zinc toxicosis can clinically manifest as a hematologic abnormality (cytopenias and coagulopathies), a gastrointestinal disorder, a cardiovascular disorder, a musculoskeletal disorder, a hepatic dysfunction, pancreatitis, renal failure, a reproductive or developmental disorder, and a neurotoxicity [13,14,15,16,17,18,19]. These manifestations are probably related to the duration of the zinc-containing object’s presence in the stomach’s acidic environment: the longer the object sits in the stomach, the more zinc is absorbed systemically. A prolonged and untreated exposure to zinc results in death. 

Newborns obtain zinc from their mother’s milk during lactation. Accordingly, one of the consequences of zinc toxicosis in a lactating NHP is zinc toxicosis in her nursing infants. Overconsumption of dietary zinc is often associated with copper deficiency and results in a zinc–copper imbalance. This imbalance has been observed in both captive and wild nursing offspring and contributes to a clinical syndrome that is known as “white monkey syndrome” (WMS). WMS is characterized by alopecia, cachexia, dermatitis, diarrhea, dehydration, emaciation, and whitening of hair (blond color of the fur), skin, and mucous membranes [13,14,15]. 

The diagnosis of zinc toxicosis is based on clinical presentation, laboratory testing (serum zinc concentration), and abdominal imaging to detect the presence of a metal-dense foreign object in the gastrointestinal tract [19]. If a metal object is not detected in the gastrointestinal tract, zinc toxicosis is often attributed to another etiology because the clinical signs are not specific [14,16,17,18]. 

Treatment of zinc toxicosis comprises relocation, removal of the zinc source, and supportive care, which can include a proton pump inhibitor (omeprazole), a gastroprotectant (sucralfate), blood products, and fluids. 

## 4. Phytotoxins

Phytotoxins are natural compounds that are inherently abundant in plants and are synthesized by plants to protect themselves against predators, insects, and microorganisms or in response to environmental stress. Phytotoxins can also underlie a plant’s characteristics, such as its aroma, color, and flavor. The chemical or molecular structure of phytotoxins is diverse and encompasses alkaloids, terpenes, phenylpropanoids, and polyketoids. Poisonous plants are classified according to the chemical nature of their phytotoxin(s), their phylogenetic relationships, or their botanical characteristics. 

Although NHP are omnivores, they spend most of their waking time foraging and searching for and processing food. Phytotoxin poisoning in NHP occurs after ingesting plants that are not intended for eating. Accordingly, NHP can be poisoned after ingesting a toxic plant because (a) hunger may cause them to graze a plant that would not be eaten under normal circumstances, or (b) they are undernourished and ingest a toxic plant to provide caloric intake. The consequences on the health of NHP after ingesting a toxic plant can range from none to sudden death. This large in-between range of adverse consequences comprises clinical signs of liver or kidney damage, cardiovascular, nervous system, musculoskeletal or gastrointestinal problems, allergic reactions, birth defects, and reproductive failure. 

Most reports of phytotoxin poisoning describe such poisonings in captive NHP (Table 2). Descriptions of phytotoxin poisoning in wild NHP have not been reported because they have developed strategies to cope or avoid this toxic threat, namely avoidance, dilution, gastrointestinal degradation, or detoxification. For example, Zanzibar red colobus monkeys (*Procolobus kirkii*) eat the charcoal from burnt trees. This behavior is thought to be a learned behavior for counteracting the toxicity of phenols in the Indian almond and mango leaves, which constitute a major part of their diet [20]. Black-and-white colobus monkeys (*Colobus guereza*) frequently come to ground to eat water plants and clay [21]. Although water–plant consumption may remedy mineral deficiencies [22], the clay may be consumed to adsorb plant toxins (usually noxious phenols) and promote their excretion, in addition to adjusting the pH of the forestomach. 

Wild NHP consume many types of plant materials that are native to the regions in which they live. This plant material, referred to as browse, is an important part of the diet of captive NHP. To date, little research has been conducted to substantiate the safety of browse species for captive NHP. The National Research Council of the National Academies has published a list of NHP-safe plants that can be used for browse [23]. However, the authors of the list do not guarantee the safety of ingesting these plants. 

Table 2 summarizes the published reports of unintentional phytotoxin poisoning in captive NHP. No antidote or specific treatments for the poisonings were provided because the poisonings were not recognized before death.

## 5. Pesticides

A pesticide is a naturally occurring or synthetic chemical or biological agent that is used to control pests. About 700 pesticides are in current use and these pesticides can be classified according to (a) the type of target pest (organism), such as a herbicide, an insecticide, a nematicide, a rodenticide, a fungicide, and a bactericide; (b) the chemical structure, such as organophosphates, organochlorines, carbamates, and pyrethroids; and (c) the mechanism (mode) of action, such as enzyme inhibitors, disruptors of cellular signaling pathways, and generators of reactive molecules that destroy cellular components, amongst others. To be acceptable, a pesticide must be toxic to the intended target and not toxic to any non-target organism. 

Wilderness areas are vital refuges where natural ecological and evolutionary processes can operate with minimal human disturbance. Humans have appropriated much land and altered terrestrial ecosystems for agriculture. Agricultural frontiers are dynamic environments that are characterized by the conversion of native habitats to agriculture and the highest incidence of species loss occurs on an agricultural frontier. Therefore, habitat loss is a burgeoning threat to a population of wild NHP and other terrestrial vertebrates that live on an agricultural frontier. Furthermore, the use of pesticides on an agricultural frontier has the potential to detrimentally affect wild NHP populations, the local biodiversity, and ecosystem’s structure and function. 

An overview of the reported cases of pesticide poisonings in wild NHP are presented in Table 3A,B. Since these poisonings occurred mostly on agricultural frontiers, pesticides, which were used to control agricultural pests, were the suspected cause of these poisonings in non-target wild NHP. The main reported outcome of an acute poisoning by a pesticide was death (Table 3A). Continual exposure to a pesticide on an agricultural frontier may also be hazardous for wild NHP populations because of the possible teratogenic effects of the pesticide: birth (congenital) defects have been reported in the offspring of pesticide-exposed pregnant mothers (Table 3B). These effects were usually documented in the reports of research projects whose aims were to (a) determine the cause of the phenotypical harm to wild NHP and (b) understand whether these harms were linked to environmental pollution of their habitat. Unfortunately, no definitive links between pesticide exposure and reproduction or frequency of stillbirths could be made. Interestingly, many health authorities advise pregnant women or women who are planning to become pregnant avoid contact with or exposure to pesticides because some pesticides may cross the placental barrier and lead to miscarriages, pre-term births, infants with low birth weights, birth defects, and congenital anomalies. 

## 6. Botulism

Botulinum toxins are neurotoxins that are produced by spores of the anaerobic bacterium, *Clostridium botulinum*. There are seven botulinum toxins, namely, A, B, C, D, E, F, and G. Types A, B, E, and F cause botulism in humans and types C, D, and E are the causes of botulism in other mammals and vertebrates. Since the spores are resistant to heat, botulism usually occurs after the ingestion of improperly processed food or spore-contaminated water (food-borne and water-borne botulism). However, botulism can also occur when a wound becomes infected with the spores or inhalation of the spores. The main clinical signs of food-borne botulism are an initial ataxia, which is followed by flaccid paralysis of the legs and neck muscles and respiratory depression and failure [51]. Botulism is a rarely reported disease in NHP. The published reports on botulism in captive NHP indicate that the cases were food- and water-borne botulism caused by the type C toxin [52,53,54]. Diagnosis is usually based on the history and clinical signs, and laboratory confirmation of the toxin’s presence in the suspected source and blood, body fluids, and excretions of affected animals. Treatment comprises administration of antitoxin, mechanical ventilation, and other supportive treatments. Untreated botulism is fatal.

## 7. Polychlorinated Biphenyls

Polychlorinated biphenyls (PCBs) are human-made chlorinated organic chemicals and were widely used in innumerable industrial and commercial applications. Since PCBs are chemically stable and non-inflammable and have electrical insulating properties, they (a) were present in electrical transfer and hydraulic equipment, (b) used as plasticizers in paints and rubber products, and (c) constituents of pigments, dyes, and carbonless copy paper, amongst others. PCBs were manufactured worldwide from 1929 until the late 1970s and early 1980s, when their manufacture was terminated when authorities realized that PCBs were persistent environmental and toxic contaminants. The primary sources of environmental PCBs include their vaporization after undisclosed uses in unenclosed areas, inappropriate disposal practices, volatilization and runoff from landfills that have PCB-containing wastes, accidental release of PCBs from facilities where they are used, and incineration of PCB-containing wastes.

PCBs have become ubiquitous environmental contaminants because they do not easily degrade in the environment. PCBs can bioaccumulate in adipose tissue of NHP because they are lipophilic. PCB congeners such as polychlorinated dibenzo-p-dioxins (PCDDs) and dibenzofurans (PCDFs) are carcinogenic. 

Several reports on PCB toxicity in captive NHP have been published [55,56,57,58]. This toxicity is a chronic and progressive disease, which is characterized initially by diarrhea, weight loss, dehydration, weakness, lethargy, inappetence, followed by alopecia, acne, facial edema, swelling and reddening of the eyelids, gingivitis, and emaciation, and then death. PCBs also possess teratogenic properties. The infants of PCB-exposed mothers are small and unthrifty, and the infant mortality rate is high. Laboratory and main postmortem findings of PCB toxicity were hypertrophic and hyperplastic mucinous gastropathy and the presence of PCBs in the livers of the dead NHP [59,60,61]. The source of the PCBs was the concrete sealers that were used during building (re)construction [55,56,57,58]. The treatment comprised the removal of the PCB-containing sealer and resurfacing the floors of the NHP’s housing facility.

## 8. Snake and Spider Venoms

Snakes are ectothermic limbless reptiles that regulate and maintain their body temperature by relying on the environment. Venomous snakes are found in most parts of the world except in very cold regions and many islands and have been identified in only five families: *Elapidae*, *Hydrophiidae*, *Viperidae*, *Crotalidae*, and *Colubridae*. Venom is produced and secreted by the snake’s oral glands in order to immobilize and kill its prey and aid in its digestion. The composition of snake venom has a large degree of variability: enzymes and proteins of various sizes (phospholipases, serine proteases, catalase, hyaluronidase, and collagenase), amines, lipids, nucleosides, and carbohydrates. Venoms act on many cell types and their actions include cell and cell membrane digestion, disruption of the functional role of the procoagulant and anticoagulant systems in blood, production of reactive oxidizing agents, breakdown of collagen and the intercellular matrix, and neurotoxicity. Venoms can also be classified according to their main toxin, namely a neurotoxin, a hemotoxin, a cardiotoxin, a cytotoxin, or a myotoxin and the target organ because they cause paralysis, coagulopathy, rhabdomyolysis, and organ failure.

There are only a few published reports of (fatal) attacks on wild NHP by venomous snakes, usually terrestrial vipers [62,63,64,65]. A bite from a *Crotalus adamanteus* has immediate local and systemic effects. The local effects at or near the site of the bite include marked edema (and apparently marked pain), followed by rupture of the skin, and tissue and muscle necrosis. The systemic effects are dependent upon the venom’s composition and confined to general lethargy, salivation, bloody diarrhea, and death in an irreversible coma. Hydrocortisone treatment is probably therapeutically beneficial for NHP following a venomous snake bite [66].

Spider venoms are a mixture of many active compounds, some of which have cytolytic or neurotoxic activity. The Australian funnel-web spider, which has been designated as the world’s deadliest spider, is named so because its venom contains a lethal neurotoxic polypeptide, robustoxin [67]. This neurotoxin is a δ-hexatoxin (δ-HXTXs) and this family of toxins is responsible for the human envenomation syndrome. In human bite victims, δ-HXTXs cause disturbances in respiration, blood pressure, and heart rate, followed by severe hypotension. Without treatment with commercial antivenom, fatalities can occur by respiratory and circulatory failure within a few hours of the bite. Wiener [68] reported on the death of and the main post-mortem findings in a captive cynomolgus monkey (*Macaca fascicularis*) that died within 36 h after being bitten by a funnel-web spider. The main post-mortem findings were areas of emphysema, hemorrhage, and edema in the lungs. Although Gray and Sutherland [69] reported that a captive monkey may sometimes die after being bitten by a female funnel-web spider, there is no certainty that venom has been injected after a spider bite [68]. It has also been reported that captive cynomolgus monkeys can be protected against the lethal effects of male funnel-web spider venom by immunization with robustoxin glutaraldehyde-polymerized toxoid [70,71] or a synthetic robustoxin derivative lacking disulfide bridges [72]. 

## 9. Miscellaneous

Toxic threats to captive NHP differ because of the environment in which they live. These threats include ubiquitous environmental pollutants, cleaning solvents [73], veterinary medicines [74,75], and constituents of the components of a building or an enclosure structure [76]. Although several authors have reported single cases of a presumed intoxication, these reports often lack details of the poison’s source, the NHP’s clinical signs, the laboratory and any postmortem findings, and treatment [77,78,79].

## 10. Summary and Conclusions

In this review, we have tried to make the reader aware that (a) NHP are exposed and potentially vulnerable to many man-made toxic threats and (b) the toxic threats to wild NHP are different to those to captive NHP because they live in dissimilar environments. For wild NHP, the predominant toxic threat is that of pesticides because of their application in agricultural frontiers where NHP are one the many non-target species that inhabit these frontiers. Although the other toxic threats to wild NHP, namely phytotoxins and the venoms of predatory terrestrial snakes, spiders, and probably insects, are minor and weak, all are potentially lethal. The housing environments of captive NHP are varied. A captive NHP may be housed in (a) a facility (vivarium) attached to or affiliated with a research institute because the NHP is used as a research animal; (b) a zoo or sanctuary whose purpose is to preserve a species and educate the public in order to foster an appreciation of other species; and (c) a family home because the NHP is kept as a domestic pet. Therefore, the toxic threats to captive NHP are also varied because they also live in dissimilar environments. These threats include ubiquitous environmental pollutants, cleaning solvents, and constituents of the components of a building or enclosure structure. These findings and conclusions were based on a critical examination of published reports that were mined from academic literature databases. We recognize that these reports probably represent only a fraction of the number of intoxications that may have occurred in wild and captive NHP. We also emphasize that it is often unclear in some reports that the clinical signs and the results of laboratory testing and post-mortem examinations can be exclusively attributed to a specific toxic agent. Therefore, we recommend that a poisoning or an intoxication should be part of a differential diagnosis when a NHP presents with clinical signs that cannot be linked to a known disorder or dies suddenly with no preceding clinical signs. In order to authenticate a diagnosis of a poisoning in cases of the former, laboratory testing for known toxins and pollutants should be conducted. In order to authenticate a diagnosis of poisoning in cases of the latter, a thorough postmortem examination, together with extensive laboratory testing, for known toxins and pollutants in all tissues and organs should be conducted on all dead NHP.

## Figures and Tables

**Table 1 animals-12-00854-t001:** The main toxic agents in nonhuman primates.

Section	Toxic Agent
2	Lead
3	Zinc
4	Phytotoxins
5	Pesticides
6	Botulinum toxin
7	Polychlorinated biphenyls
8	Snake and spider venoms
9	Miscellaneous

**Table 2 animals-12-00854-t002:** Cases of unintentional phytotoxin poisoning in captive NHP.

NHP Species	Plant Species	Phytotoxin	Clinical Signs	Clinical Signs	Reference
*Cebus apella*	English ivy (*Hedra helix*)		Severe gastroenteritis resulting in acute death	Not performed	Fowler, 1981 [24]
White mantled-black colobus (*Colobus guereza*)	Alleghany (*Viburnum x rhytidophylloides**)*		Pain, lethargy, inappetence, unable to climb, vomiting, diarrhea resulting in death	Small spicules of plant material were present in the inflammatory exudate associated with the stomach’s mucosa	Irlbeck et al., 2001 [25]
François’ langurs (*Trachypithecus francoisi)*	Hybrid yew shrub (*Taxus baccata X T. cuspidata*)	Taxine alkaloids	Found dead without previous clinical signs	Multiple yew fragments in gastric content and taxine alkaloids were detected by gas chromatography and mass spectrometry of the gastric contents	Lacasse et al., 2007 [26]
Alaotran gentle lemur (*Hapalemur griseus alaotrensis*)	Russian vine (*Polygonum baldschuanicum*)	Oxalate	Varying degrees of lethargy, inappetence, abdominal discomfort and diarrhea progressing to signs of renal insufficiency (hematuria, proteinuria, and severe uremia) resulting in death	Chronic renal failure, presence of calcium oxalate crystals in the renal tubules	Scott, 1996 [27]
black and white ruffed lemurs (*Varecia Variegate Variegate)*	Hairy nightshade (*Solanum sarrachoides*)	Alkaloid glycoside	Acute death or less active and partial inappetence for 48 h followed by depression, lethargy, ataxia, diarrhea, and slow pupillary reflexes prior to death	acute, severe, diffuse hemorrhagic enteritis and typhlitis and a small volume of unidentifiable plant material was found in the stomach contents	Drew & Fowler, 1991 [28]

**Table 3 animals-12-00854-t003:** (**A**) Summary of the reported cases of death in wild NHP due to acute pesticide poisoning on agricultural frontiers. (**B**) Summary of the reported birth defects and congenital anomalies in wild NHP due to suspected continual pesticide exposure on agricultural frontiers. * The clinical signs and main laboratory findings and pathology were often not performed and documented in the report.

(**A**)				
**NHP Species**	**Suspected Pesticide**	**Clinical Signs**	**Laboratory and Main Postmortem Findings**	**Reference**
Not specified	Anticoagulant rodenticide	Not provided	Not performed	Bates, 2016 [29]
Vervet monkeys (Chlorocebus pygerythrus)	Aldicarb, carbofuran	Sudden death	Not performed	Botha et al., 2015 [30]
Golden Langurs (*Trachypithecus geei*)	Organochlorine insecticide	Sudden death	Insecticide was detected in the liver, kidney, and intestinal contents	Pathak, 2011 [31]
Bonnet macaque (*Macaca radiata*)	Carbofuran (a carbamate insecticide)	Sudden death	Cyanosis, severe pulmonary congestion, splenomegaly and dark purplish-blue granules, identified as carbofuran, in the gastric contents	Radhakrishnan, 2017, 2018 [32,33]
Cynomolgus monkeys (*Macaca fascicularis*)	Anticoagulant bromadiolone and difenacoum	Sudden death	extensive subcutaneous and internal hemorrhages. Bromadiolone and difenacoum were detected in frozen liver samples	IJzer et al., 2009 [34]
Squirrel Monkey (*Saimiri sciureus*)	Fipronil	Ranging from sudden death to symptoms of depression, inappetence, lethargy and body weight loss, which progressively disappear over time	Fipronil and fipronil sulfone were detected in cutaneous and brain tissue	Demir et al., 2021 [35]
Tantalus monkeys (*Cercopithecus aethiops*)	Dieldrin	Sudden death	Not performed	Koeman et al., 1978 [36]
**(B)**				
**NHP Species**	**Suspected Pesticide**	**Clinical Signs**	**Laboratory and Main Postmortem Findings**	**Reference**
Ring-tailed lemurs	Organochlorine pesticides	*	*	Rainwater et al., 2009 [37]; Dutton et al., 2003 [38]; Miller et al., 2007 [39]
Chimpanzees and baboons	Several different pesticides	(Congenital) facial and nasal deformities (i.e., reduced nostrils, cleft lip), limb deformities, reproductive problems, and hypopigmentation	*	Krief et al., 2017 [40]; Lacroux et al., 2019 [41]
Baboons, howler monkeys, chimpanzees, red-tailed monkeys, red colobus	Pesticides, halogenated flame retardants, and organophosphate flame retardants	*	*	Wang et al., 2020 [42]
Douc langurs (*Pygathrix* spp.)	Dioxins (i.e., Agent Orange, tetrachlorodibenzo-p-dioxin) and dioxin-related compounds	Two animals exhibited developmental consequences of possible dioxin exposure	*	Brockman et al., 2009 [43]; Brockman & Harrison, 2013 [44]
Baboons (*Papio* spp.), Tantalus monkeys (*Chlorocebus tantalus*), red tail monkeys (*Cercopithecus ascanius*), vervet monkeys (*C. pygerythrus*), Campbell’s monkeys (*Cercopithecus campbelli lowei*), Zanzibar red colobus (*Procolobus kirkii*), and chimpanzees (*Pan troglodytes*)	Pesticides	*	*	Ogada, 2014 [45]; Naughton-Treves, 1998 [46]; Eniang et al., 2011 [47]; Nowak et al., 2009 [48]; Sai et al., 2006 [49]
Japanese monkeys (*Macaca fuscata*)	Dieldrin and heptacholorepoxide	Congenital defects such as abnormal limbs in offspring	Elevated concentration of dieldrin and heptacholorepoxide in the liver and kidney of female monkeys whose babies were born with malformations	Minezawa et al., 1990 [50]

## Data Availability

Not applicable.

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
