# Peer review of "A Literature Review of Unintentional Intoxications of Nonhuman Primates"

_animals, 2022, doi:10.3390/ani12070854_

Round 1

Reviewer 1 Report

The topic of the paper is to provide a review of unintentional intoxications of NHP both in the wild and in captivity. Most intoxications are reported as case reports and an overview of the poisoning is needed. for this reason, this article would be a useful contribution.

Few general comments :

  • the report is mostly on captive NHP
  • the report is mostly on accute intoxication

It would be interesting to explain why

Subtitles would be useful for the structure of the paper and some information would be useful to be provided such as frequency of the cases and dose of exposure leading to intoxication (accute or chronic).

  • sources of contamination
  • way of exposure, dose of exposure for -letal- intoxication
  • symptoms
  • treatment

It would be good to have an estimation of the number of reported cases in the litterature for each group of toxic described.

Specific comments :

simple abstract : objectives of this review are lacking. the last sentence is really weird : coudl we really call "imporvement"the practices which expose to death?

Abstract : it could be important to specify most cases are "accute" intoxication

l56 the following sentence is not clear; "these guidelines do not adress the location of a housing system" I guess this is related to l 45-47?

In the introduction and in the discussion, paragraphs related to wild NHP would be necessary.

Author Response

Dear reviewer 1,

We would like to thank you for your comments and remarks. We have revised the manuscript according to your remarks. Please find below our point-to-point reply:

Comment #1: The reviewer commented that the review focuses on captive NHP.

Response: Although we agree with the reviewer, we feel that including the word “captive” in the review’s title is potentially misleading because the review discusses pesticide poisoning and the effect of snake bites in wild NHPs.

Comment #2: The reviewer commented that the review focuses on acute intoxications in NHP.

Response: We do not agree with this comment. For example, lead poisoning and zinc toxicosis are not acute intoxications but chronic poisonings whose clinical signs are associated with exposure time-dependent accumulation of the metals. Therefore, we feel that including the word “acute” in the review’s title is inappropriate.

Comment #3: The reviewer requested that the review include information on the source of the poison, the method of exposure, the toxin’s dose, the symptoms and treatment of each poison, and the reported number of poisonings.

Response: We have provided the missing information where possible and all missing references are included in the revised version.

Comment #4: The reviewer commented on the messages of the last sentences of the simple summary (lines 13-16).

Response: We have revised the text of these sentences so that their message is coherent.

Comment #5: The review requested that the abstract includes acute intoxications.

Response: See our response to comment #2. We do not agree with this comment. For example, lead poisoning and zinc toxicosis are not acute intoxications but chronic poisonings whose clinical signs are associated with exposure time-dependent accumulation of the metals. Therefore, we feel that including the word “acute” in the review’s title is inappropriate.

Comment #6: The reviewer commented on the sentence on line 57 “Remarkably, these guidelines do not address the location of a housing system for NHP.”

Response: This sentence has been deleted in the revised version.

Comment #7: The reviewer requested that the introduction and discussion distinguish between captive and wild NHP.

Response: We have added the words “captive” and “wild” to the text where we considered that it was needed to distinguish between captive and wild NHP.

Kind regards,

Jaco Bakker

Reviewer 2 Report

Journal: Animals

Manuscript Title: A Literature Review of Unintentional Intoxications of Nonhuman Primates

Manuscript Number: animals-1604030-peer-review-v1

The author reviews unintentional intoxications of nonhuman primates. The review is meaningful. However, the authors are suggested to revise the review before being published in “Animals”. My comments are as follows.

  1. L 23: Revise“using words” to “used words”.
  2. L 25: Revise“according to the” to “according to”.
  3. L 28: “innumerable man-made toxic threats”?
  4. There is a lot of repetition in the text and abstract
  5. A lot of information unrelated to poisoning is involved in
  6. L 70: Revise“we describe” to “we described”.
  7. Many of the contents in the manuscript do not provide references, such as “Zinc toxicosis mainly occurs in captive NHP and is usually the result of licking the galvanized metal of their cages or enclosures or the ingestion of galvanized nuts and bolts from their metal cages or enclosures”on line 130-132.

Author Response

Dear reviewer 2,

We would like to thank you for your comments and remarks. We have revised the manuscript according to your remarks. Please find below our point-to-point reply:

Comment #1: The reviewer recommended that “using words” on line 23 be changed to “used words”.

Response: Since “using” is a gerund, we have inserted the word “the” so the text will read “using the words”.

Comment #2: The reviewer recommended that “according to the” on line 25 be changed to “according to”.

Response: We have not made the recommended change because the word “the” is used as a definite article in order to identify the noun, namely “agent”.

Comment #3: The reviewer commented on the sentence on lines 27-28 “NHP are exposed and potentially vulnerable to innumerable man-made toxic threats.

Response: This sentence has been deleted in the revised version.

Comment #4: The reviewer commented on repetition and the presence of unrelated information in the text.

Response: We have critically reviewed the text of the revised version and we hope that its text now meets the requirements of the reviewer.

Comment #5: The reviewer recommended that “we describe” on line 70 be changed to “we described”.

Response: We have not made the recommended change because a reader is reading our descriptions in the review. Accordingly, the sentence’s tense has been written in the present tense.

Comment #6: The reviewer requested that the authors provide references for the sentence “Zinc toxicosis mainly occurs in captive NHP and is usually the result of licking the galvanized metal of their cages or enclosures or the ingestion of galvanized nuts and bolts from their metal cages or enclosures” on lines 130-132 and other unidentified sentences.

Response: All missing references are included in the revised version.

Kind regards,

Jaco Bakker

Reviewer 3 Report

Please consider supplementing the study with the issue of mycotoxin contamination.

I recommend literature:

Stuper K., Cegielska-Radziejewska R., Szablewski T., Ostrowska A., Busko M., Perkowski J. 2015. Contamination with microscopic fungi and their metabolites in chicken feeds produced in Western Poland in the year 2009-2010. Acta Scientarum Polonorum. Zootechnica. 14(3), 107-122.

Author Response

Dear reviewer 3,

We would like to thank you for your comments and remarks. We have revised the manuscript according to your remarks where possible. Please find below our point-to-point reply:

Comment #1: The reviewer recommended that the authors consider referring to the published report “Stuper K., Cegielska-Radziejewska R., Szablewski T., Ostrowska A., Busko M., Perkowski J. 2015. Contamination with microscopic fungi and their metabolites in chicken feeds produced in Western Poland in the year 2009-2010. Acta Scientarum Polonorum. Zootechnica. 14(3), 107-122” in a revised version.

Response: We have not included this report in the revised version because we found no published reports on poisoning of captive NHP due to the ingestion of a commercially available diet which is given to captive NHP. 

Kind regards,

Jaco Bakker
